# Biofunctionality of Enzymatically Derived Peptides from Codfish (*Gadus morhua*) Frame: Bulk In Vitro Properties, Quantitative Proteomics, and Bioinformatic Prediction

**DOI:** 10.3390/md18120599

**Published:** 2020-11-27

**Authors:** Ali Jafarpour, Simon Gregersen, Rocio Marciel Gomes, Paolo Marcatili, Tobias Hegelund Olsen, Charlotte Jacobsen, Michael Toft Overgaard, Ann-Dorit Moltke Sørensen

**Affiliations:** 1Research Group for Bioactives-Analysis and Application, Division of Food Technology, National Food Institute, Technical University of Denmark, 2800 Kongens Lyngby, Denmark; s182389@student.dtu.dk (R.M.G.); chja@food.dtu.dk (C.J.); adms@food.dtu.dk (A.-D.M.S.); 2Section for Biotechnology, Department of Chemistry and Bioscience, Aalborg University, 9220 Aalborg, Denmark; mto@bio.aau.dk; 3Department of Health Technology, Technical University of Denmark, 2800 Kongens Lyngby, Denmark; pamar@dtu.dk (P.M.); tobhe@dtu.dk (T.H.O.)

**Keywords:** codfish, enzymatic hydrolysis, proteomics, bioinformatic prediction, emulsifying properties, antioxidative activity, bioactive peptides

## Abstract

Protein hydrolysates show great promise as bioactive food and feed ingredients and for valorization of side-streams from e.g., the fish processing industry. We present a novel approach for hydrolysate characterization that utilizes proteomics data for calculation of weighted mean peptide properties (length, molecular weight, and charge) and peptide-level abundance estimation. Using a novel bioinformatic approach for subsequent prediction of biofunctional properties of identified peptides, we are able to provide an unprecedented, in-depth characterization. The study further characterizes bulk emulsifying, foaming, and in vitro antioxidative properties of enzymatic hydrolysates derived from cod frame by application of Alcalase and Neutrase, individually and sequentially, as well as the influence of heat pre-treatment. All hydrolysates displayed comparable or higher emulsifying activity and stability than sodium caseinate. Heat-treatment significantly increased stability but showed a negative effect on the activity and degree of hydrolysis. Lower degrees of hydrolysis resulted in significantly higher chelating activity, while the opposite was observed for radical scavenging activity. Combining peptide abundance with bioinformatic prediction, we identified several peptides that are likely linked to the observed differences in bulk emulsifying properties. The study highlights the prospects of applying proteomics and bioinformatics for hydrolysate characterization and in food protein science.

## 1. Introduction

Approximately 60% of fish processing byproducts are discarded in the sea or used as fishmeal, fish oil, fertilizer, fish silage, and animal feed [1]. Fish frame is the major side stream from fish processing plants, which comprise about 22% solids and considerable amounts of proteins and minerals. This means high disposal costs for the processors and a negative value for the industry. However, the increasing demand for protein on a global scale turns the focus on underutilized protein sources such as cod (*Gadus morhua*) frame. Cod frame side-streams can be considered as a steady and valuable source of not only minerals such as calcium and phosphorus, but also protein and potentially bioactive peptides (BAPs) [2]. Cheung et al. reviewed recent studies on the utilization of marine species by enzymatic hydrolysis for the recovery of various valuable components [3]. For instance, fish protein hydrolysates (FPH) from cod frames have been shown to possess functional and antioxidant activities [4,5,6].

Proteolytic conversion of proteins into bioactive peptides has become an integral part of the food processing industry and has many advantages. The creation of new nutraceuticals and functional food ingredients can promote the health level of society and/or improve the functional and rheological properties of food products [7]. The application of proteolytic enzymes has been broadly considered as the most practical approach for the release of BAPs from fish protein [8,9]. To obtain highly functional BAPs, it is necessary to optimize the hydrolyzation process [10]. The use of different types of proteases, substrates (different fish species/flesh/frame/viscera), and several other variables such as pH, time, and temperature have been extensively investigated [11].

Studies on identification of the generated peptides from fish proteins as a function of the proteases type are scarce [12,13] and there is no report specifically in the case of cod frame. Consequently, this study was performed to increase the knowledge of practical short-term hydrolysis of cod frame by two industrial proteases: Alcalase and Neutrase, either individually or sequentially. The efficiency of the selected proteases was assessed in terms of the degree of hydrolysis (DH), and the composition of hydrolysates was characterized using mass spectrometry (MS)-based proteomics. The obtained hydrolysates were characterized with respect to their emulsifying and antioxidative properties. In addition, bioinformatics tools were used to predict the emulsifying and antioxidative properties of the most abundant peptides from cod frame, as determined by MS. Combining the application of proteomics and bioinformatics tools has been suggested to be an effective approach in early stage identification of embedded bioactive peptides in order to design a protocol for hydrolysis (e.g., using enzymes with the appropriate specificity) and further purification of the hydrolysate, saving costs and time of extensive screening processes [14]. Thus, by combining quasi-quantitative identification of peptides with predicted functionality using bioinformatics, we aimed to identify the BAPs responsible for the bulk properties observed in the FPHs.

## 2. Results and Discussion

### 2.1. Proteolysis Efficiency

The degree of hydrolysis (i.e., percentage of cleaved peptide bonds) is a true reflection of the progress of hydrolysis and thus its selection as the primary indicator for controlling or measuring the extent of hydrolysis is very appropriate. As shown in Table 1, the application of proteases, either individually or sequentially applied to cod frame, resulted in DH between about 19% and 39% (*p* < 0.05). The highest DH was obtained by sequential enzyme treatment of MCF while the lowest DH was obtained with Neut on HCM. The measured DH range corresponds well with previous reports on short-term hydrolysis of cod frame using Protamex. Alc hydrolysis on both MCF and HCM substrates showed higher efficiency i.e., higher DH compared to Neut hydrolysis. This is confirmed by both tricine SDS-PAGE analysis in Figure 1 and gradient SDS-PAGE analysis (Appendix A), where Neut hydrolysates (lanes 2 and 5) show a larger proportion of higher more molecular weight (MW) peptides compared to both Alc (lanes 1 and 4) and sequential (lanes 3 and 6) hydrolysates. When looking at SDS-PAGE staining, it should be kept in mind that response is mass-dependent and hence not stochastic. This means that, in principle, there has to be ten times more of the number of e.g., a 500 Da peptide to produce the same response as one equivalent of a 5 kDa peptide. The observation is not surprising though, as Alc (which mainly consist of subtilisin A) shows very broad specificity towards bulky and uncharged amino acids (AAs) [15].

Neut, on the other hand, mainly prefers leucine and phenylalanine specifically [15]. As both AAs are bulky and uncharged, they are also targeted by Alc. Higher activity of Alc compared to Neut at the same E/S ratio is also in line with other studies [16]. The overlap in specificity may also explain why very low increase in DH is observed for sequential hydrolysis of MCF, while no significant increase in DH is observed for HCM (Table 1). Based on DH, both proteases display substrate preference towards MCF, an observation not readily observable from SDS-PAGE analysis (Figure 1). Considering that HCM is a heat-denatured substrate, formation of protein aggregates may decrease accessibility thereby decreasing DH for HCM. Although the boiling of the cod frame was done well below the decomposition temperatures of Phe (>200 °C) [17], thermally induced oxidation of aromatic AAs, hereunder Phe, is a well-documented phenomenon [18]. Furthermore, Phe has also been reported to trigger protein aggregation at elevated temperatures [19,20]. Consequently, as Phe and other thermally oxidized AAs are altered or become inaccessible from heating, it is speculated that the number of proteolytic sites have been reduced. This also explains why the effect of boiling is more pronounced for Neu than Alc (relative decrease in DH of 30% and 16%, respectively). When considering the AA composition of the cod frame substrate, the relative decrease of 30% in DH for Neut corresponds perfectly with the amount of Phe relative to the amount of Phe and Leu combined [2].

To make the hydrolysis process more industrially and economically applicable, the enzyme was added based on its weight relative to the protein concentration of the initial substrate, and not based on declared enzymatic activity per enzyme mass. This means that Alc was applied in a three times higher activity than Neut (2.4 AU/g vs. 0.8 AU/g, respectively). This factor should also be regarded when comparing hydrolysis efficiency, and, therefore, the prerequisite for proper comparison between these two enzymes is to administrate them with the same activity as done in a study conducted by Liaset et al. [21]. The authors reported that by application of Alc and Nuet at 30 AU/g of cod backbone, no significant differences were observed in terms of DH after 60 and 120 min of enzymatic hydrolysis process. Nevertheless, the difference in specificity will also be reflected in the resulting hydrolysate and consequently also the properties hereof.

When comparing DH between studies as the main variable, several factors must be considered. Apart from enzyme specificity, variation in DH can be attributed to other experimental variables such as substrate types, testing conditions (time, temperature, pH, E/S ratio, etc.), seasonal variation and catchment region [22]. In addition, evaluating the hydrolysis efficiency by DH is done in fundamentally different ways, which will influence the result [23]. Amongst the most popular are pH stat [24], soluble nitrogen in aqueous trichloroacetic acid (SN-TCA) [25], 2,4,6-trinitrobenzene 1-sulphonic acid (TNBS) [26], o-phthaldialdehyde (OPA) [27] and the most recent quantitative colorimetric free α-amino-N by PicoEXPLORERTM, which was used in the current study.

### 2.2. Proteomics Analysis and Methodological Limitations

The distribution of peptide chain length (PCL), peptide molecular weight (PMW), and peptide charge (Pz) is illustrated in Figure 2. Peptide properties are presented as both weighted distributions according to their relative MS1 intensities and as unweighted distributions. Binned distributions are illustrated in Appendix A. Mean population properties (weighted and unweighted) are presented in Table 2. As a direct consequence of the broad distributions of all three peptide parameters and a resultantly large standard deviation, there are no statistically significant difference between means at a 95% confidence level using Tukey HSD. Nevertheless, both the mean properties (Table 2) and the distributions differ. Although PCL and PMW are not directly proportional, as PMW depends on the specific AA composition of a peptide, they follow the same trend. For PCLavg, the trend follows the trend observed from DH determination and the mean length based hereon (PCL_DH_) as well as SDS-PAGE analysis (Figure 1), where Alc&Neut was the most and Neut the least efficient.

Comparing PCL_avg_ and PCL_DH_ directly (Table 2), we observe a highly significant difference, as PCL_avg_ is roughly 4-fold higher than PCL_DH_ in all cases. Besides the methodological influence for determination of DH discussed above, there are several factors, which could explain this significant difference. PCL_DH_ is by definition an approximation, which does not take into account the protein composition, and in particular the length distribution of the substrate. Furthermore, ninhydrin-based methods are susceptible to interferences, which can lead to overestimation of DH and hence underestimation of PCL_DH_ [24]. Such interferences may be reactive contaminants/residuals [29], contribution of N-termini, in addition to dibasic amino acids, (Lys and Arg) [30] and Pro [31] contributing through their side-chain amino group. Finally, as DH is based on the α-amino-N content in the supernatant following hydrolysis, heat inactivation, and centrifugation, it does not factor in the protein/peptide content of the sediment, and hence does not reflect the true DH of the substrate [22]. On the other hand, PCL_avg_ may also be overestimated. Flyability in ESI-MS proteomics describes the ability/probability of peptides to be ionized in the gas phase and subsequently detected. Several studies have investigated the underlying principles governing the parameter, and computational models have been produced to estimate the highly complex and sequence dependent parameter for prediction of high responding tryptic peptides as biomarkers in protein quantification [32,33,34,35]. Although peptide length in general is negatively correlated with flyability thereby introducing an intensity bias towards shorter peptides, this is not reflected in our MS data, when compared to PCL_DH_. On the contrary, our data appear to show bias towards longer peptides in all cases and regardless of intensity-weighting. Although binned data representation (Appendix A) shows a somewhat normally distributed population of PCL, unbinned data (Figure 2) show a quite different distribution. This is particularly evident from the unweighted distribution, i.e., the number of identified peptides at each PCL. Here, we see a somewhat binominal distribution with one part centered around 12–13 AAs, decreasing towards 6 AAs, and then presenting a huge spike at 3–4 AAs. In order to explain this observation, it is crucial to consider the peptide combinatorial space as a function of PCL and the computational methods used for peptide identification and subsequent quantification.

MaxQuant [36,37] employs the Andromeda search engine for peptide identification [38]. In the process of peptide identification, Andromeda calculates a peptide score based on the MS2 peptide spectrum match (PSM) with a theoretical MS2 spectrum. The score is subsequently used to control the false discovery rate (FDR) as a function of PCL. As the FDR controls the cutoff peptide score for positive PSMs and as peptide score is furthermore influenced by both PCL and intensity [36,37,39], the length becomes a critical factor when moving towards shorter peptides. As illustrated by the MaxQuant developers [36], when decreasing the PCL towards 6 AAs, the peptide score distribution for the target library and the decoy library fully coincide. As the decoy library is more densely populated at any peptide score, this means that employing standard FDR of 1% on the peptide level will result in a very small probability of a positive PSM and hence identification of a hexapeptide (PCL = 6), despite them actually being present in the sample. Ultimately, this leads to a bias towards identification of longer peptides, which has also been reported in the literature [40,41].

Decreasing the PCL even further, the situation is somewhat different. As PCL decreases, so does the combinatorial space of possible peptides, which is shown in Appendix A for both the 20 naturally occurring AAs and including the common variable modifications employed in data analysis (oxidation of Met and N-terminal acetylation). With ~75,000 protein entries in the proteome composing on average 326 AAs, there are for instance over 2×10^7^ tetrapeptides (PCL = 4). This means that the amount of tetrapeptides cover the combinatorial space for tetrapeptides over 100-fold even after including modification, although significant redundancy is expected. This directly implies that the search space in the decoy library will be depleted, as most decoy peptides will be present in the target library, which takes priority for coinciding peptides. Consequently, the peptide score distribution will be much more densely populated and the probability of a decoy PSM will be low, thereby removing the critical peptide score and FDR constraint observed for hexapeptides. As the PCL decreases further, so does the combinatorial space, and hence the possible number of identifiable peptides, thereby explaining why the number of identified tripeptides (PCL = 3) is lower than the number of tetrapeptides. Together, the two factors describe the binominal distributions observed for particularly unweighted PCL. Consequently, a “grey zone” at lower PCL (3–10 AAs) arises, where there is likely to be a large number of missing peptide identifications. As the analysis furthermore does not include free AAs and dipeptides due to the high noise in the low *m*/*z* range and the lack of MS2 evidence to provide PSMs, this ultimately makes the method significantly biased towards longer peptides, particularly for hydrolysates with high DH. In the end, this also means that the distributions and calculated means are likely overestimated and do unfortunately not reflect the actual distribution of peptides in the sample. The “grey zone” is also reflected in the number of identified peptides (Table 2). Theoretically, the number of different peptides should increase with increasing DH until a certain point. However, we observe the opposite trend. Nevertheless, as either increasing DH results in an increasing proportion of peptides of a length in the “grey zone” or hydrolyzed to dipeptides or free AAs, the combinatorial space also decreases (Appendix A), and the observed trend may in fact be a combination of both missing identifications and combinatorial space reduction. Nevertheless, more than ¼ of the identified peptides for sequential hydrolysis are tetrapeptides (Figure 2B) for both substrates indicating a high DH in these FPHs.

Using LC-MS/MS to identify peptides in hydrolysates is far from novel [42]. In most cases, this results in identification of multiple peptides, and the most abundant by MS1 intensity are subsequently reported as likely to be the source of the observed activity. Nevertheless, protein hydrolysates are highly complex mixtures of peptides identifying the highest responder may not identify the species responsible for the observed in vitro property. Instead, bulk properties are in fact the cumulative properties of all peptides in the hydrolysate. Nevertheless, highly abundant peptides may significantly govern the bulk property. Consequently, we described the mean properties of the hydrolysate here in terms on length, MW, and charge by introducing intensity-weighted means along with stochastic means (Table 2). By applying weights, we also significantly alter the distribution (Figure 2 and Appendix A). This is done using the assumption that high responders are highly abundant, and thus describe the mean population of a hydrolysate more satisfactory. Although still not statistically significantly different due to the large standard deviations, the difference between means is also larger, thereby better reflecting the differences between treatments observed using α-amino-N and SDS-PAGE analysis. Although intrinsic, sequence-specific properties such as flyability influence the intensity of a given peptide, we believe that weighted distributions and means are better descriptors. Regardless, there remains a large methodological gap for sequence-dependent correlation of MS1 intensity and absolute abundance for non-tryptic peptides.

### 2.3. Interfacial Properties

#### 2.3.1. Bulk Emulsifying Properties

Table 1 shows the EAI and ESI of FPHs from cod frame by Neut and Alc, individually or sequentially, and sodium caseinate (SC) as the control. By comparing within group data, it is noticeable that application of Neut and Alc separately on either MCF or HCM did not cause a significant difference in emulsifying properties, while their sequential administration on the HCM increased the EAI although lowering ESI, significantly (*p* < 0.05). However, both substrates revealed significantly higher ESI compared to sodium caseinate (SC) and MCF also had significantly higher EAI than SC (*p* < 0.05). Overall, we observe a positive relation between DH and emulsifying properties, i.e., the higher DH and hence lower PCL, the greater EAI. Furthermore, emulsification properties showed a significant difference (*p* < 0.05) related to the substrate, where FPH from MCF in all cases displayed higher activity but lower stability compared to HCM. Interestingly, this observation was not expected, as extensive hydrolysis generally has been associated with decreased emulsifying properties [43]. Larger peptides are considered more efficient in stabilizing the interface, while small peptides diffuse to and adsorb faster at the interface, albeit less efficient in interfacial stabilization. Although peptide length is indeed a critical factor in O/W emulsification stability [44], peptide sequence and consequently potential amphiphilicity are of higher importance [8,45]. In addition, variations in charge and thickness of interfacial layer possess a remarkable impact on the strength and range of the steric and electrostatic interactions between emulsion droplets, respectively [46].

The observed levels for EAI and ESI are in good agreement with previous studies of FPHs, where EAI is generally reported to be in the range 25–270 m^2^/g and ESI in the range 22–48 min [8]. In comparison, hydrolysates from soy and whey protein have been reported to possess EAI in the range 2–12 m^2^/g and ESI in the range 10–40 min [47]. Consequently, FPHs possess much higher emulsifying activity without any cost in emulsion stability. In fact, the FPHs produced from cod frame here are promising for production of highly stable emulsions.

#### 2.3.2. Bulk Foaming Properties

The mechanism of protein/peptide foam formation properties is based on the steady diffusion of soluble molecules toward the air–water interface with subsequent reorientation and rapid conformational change at the interface. Thus, the formation of a viscoelastic film around each gas bubble will enhance the foam stability of the system in a pH-dependent manner [16,48]. In our study, FC appears to be inversely related to EAI. This also means that increased hydrolysis reduces FC. As observed for emulsifying properties, higher activity is overall associated with reduced stability. Neut derived FPHs showed higher FC but lower FS at neutral pH compared to the Alc derived FPHs, while sequential hydrolysis resulted in the absolute lowest activity, corresponding well with the protease specificities described already. In general, observed foaming properties are in line with the previously reported ranges for FPHs, with FC in the range 23–240% and FS in the range 20–140% [8]. In line with our study, Foh et al. [16] also reported that higher DH (MW 0.2–1 KDa) obtained by application of Alc and Flavorzyme on tilapia fish resulted in lower FC compared to Neut treatments with lower DH (MW 1–8 KDa), correlating well with other studies of FPHs, where too short peptides (i.e., too high DH) resulted in decreased foaming properties [49,50]. Although foaming is governed by interfacial properties, elasticity and intermolecular interactions are of higher importance. As PCL generally increases interaction strength and hence intra- and inter-peptide elasticity, length of peptides appears to be of higher importance for foaming than for emulsifying properties [51,52]. Consequently, amphiphilicity scores predicted in this study cannot be directly correlated to foaming properties.

#### 2.3.3. Interconnection of Bulk Interfacial Properties, Peptide Abundance, and Predicted Emulsifying Activity

The presence of both hydrophilic and hydrophobic regions in a peptide sequence is a prerequisite property of emulsifying peptides, as amphiphilicity is crucial to maintaining a stable O/W interface [53,54]. Facial amphiphilicity is related to peptides with α-helix and β-strand secondary structures oriented parallel to the O/W interface, whereas γ-peptides are potentially capable of perpendicular orientation at interface due to their hydrophobic and hydrophilic regions [44], although γ-peptides are also capable of adopting well defined secondary structures. In this study, a computational algorithm based on the Kyte–Doolittle hydrophobicity scale was used to predict the emulsifying properties of the determined peptide sequences, when projected in different possible conformations. Higher emulsifying scores imply the most probable peptide conformation at the interface for a given peptide, as this conformation will result in the largest amphiphilic vector. For instance, emulsifying scores in the order γ > α > β indicates that the peptide has higher amphiphilicity based on hydrophobic and hydrophilic regions compared to the hydrophilic/hydrophobic faces obtained when projected in α-helical or β-strand conformation. Emulsifying scores for the 20 most abundant peptides in each FPH are presented in Table 3, while the predicted emulsifying activity for the 100 most abundant peptides is found in Appendix A. An emulsifying score > 2 means that a peptide is predicted to be better than 97.5% of all 40,000 random peptides of the same length, used to normalize scores. Consequently, scores > 2 imply a high probability of being an in vitro emulsifier, and these peptides are highlighted in Table 3. Although high emulsifying scores increases the probability of a peptide being an in vitro functional emulsifier, high scores are implicative and not evidence of functionality, as demonstrated by Garcia-Moreno et al. [44]. However, it also needs to be noted that in this study that only peptides in a range of 3–65 AAs were included in the proteomics analysis. Therefore, free AAs, dipeptides, and undetected peptides in the “grey zone” discussed above, and longer peptides are not included in the predicting tool, while they are bound to influence the bulk functionality of all FPH samples. Using a similar approach based on machine learning, Feger et al. [55] recently demonstrated prediction of cell-penetrating peptide building blocks and subsequent design of axial amphiphilic peptides capable of producing nanoassemblies for use as a drug delivery system.

The unexpected negative effect of PCL on observed EAI may be explained by the actual peptide-level composition of the FPHs, when considering the predicted emulsifying properties. For instance, Neut&Alc treatment resulted in high EAI in both substrates (MCF and HCM). Both FPHs have several shared peptides with both high abundance and high emulsifying scores. In particular, two peptides predicted as γ-emulsifiers, IIAPPERKYS (γ = 3.43), and GADPEDVIVA (γ = 3.10) were highly abundant in both FPHs, constituting 3.49% and 1.43% of the total peptide MS1 intensity in MCF, while constituting 1.11% and 0.95% in HCM. The significantly higher abundance (3.1-fold and 1.5-fold, respectively) of both peptides in MCF is likely related to the higher EAI observed for the FPH from this substrate compared to FPH from HCM. Although the two highly probable γ-emulsifier peptides are of lower abundance in the HCM Neut&Alc FPH, this FPH contains three additional peptides with significant abundance (0.53–0.67%) that all have high γ-scores (2.37–2.76), which may also be partly responsible for the significant EAI observed.

Interestingly, the Alc derived FPH from MCF also displays a very high EAI insignificantly different from that of Neut&Alc on MCF (*p* > 0.05). This may be directly related to the presence of a significant amount (1.16%) of the peptide DIDIRKDLYAN, which is predicted to be a β-emulsifier with a very high probability (β = 3.65). This peptide, which has the highest predicted score of all top 20 abundant peptides across all FPHs, is not observed in the top 20 abundant peptides of the Alc derived FPH from HCM, although identified as the 285th most abundant peptide with a relative abundance of merely 0.06% (Appendix A). Both Alc derived FPHs share two predicted γ-emulsifiers (KSYELPDGQVITIG (γ = 2.64) and ELPDGQVITIG (γ = 2.07)) in significant abundance, which may also contribute to the bulk emulsifying properties. Although the Alc derived FPH from HCM contains three additional peptides with scores > 2 in the top 20 most abundant (one α and two γ), it is noteworthy that none of the top six abundant peptides have scores superseding 1.23 in any conformation. This may also be directly involved with the significantly lower (*p* < 0.05) EAI observed compared to Alc derived FPH from MCF.

Both Neut derived FPHs share three highly abundant and high scoring peptides; two α-emulsifiers (LEQQVDDLEGSLEQEKK (α = 2.18) and VQHELEEAEERADIAETQVNK (α = 2.21)) and one γ-emulsifier (IITNWDDMEK (γ = 2.95)), which may contribute to the observed emulsifying activity. However, as all three peptides are of higher abundance in the FPH from HCM, which in turn has a significantly lower EAI (*p* > 0.05), these peptides are likely not the main contributors to the observed activity. However, the FPH from MCF contain two abundant peptides which are not found in the top 20 abundant peptides in the HCM FPH. The peptide LKGTEDELDKYSEALKDAQEKLE constitutes 0.77% of the relative MS1 intensity and is predicted to be an α-emulsifier with high probability (α = 2.96), while the peptide LKGADPEDVIVAA constitutes 0.76% and is predicted to be a γ-emulsifier with high probability (γ = 2.78).

As shown here, PCL may in many cases be a good indicator of emulsifying potential, but the specific sequence and the resulting structural aspects of interfacial conformation appear more important. In line with this, previous studies have shown that physical stability of fish oil emulsion is indeed influenced by PCL in an interfacial conformation dependent manner, as α-helical peptide emulsifiers were generally better when the PCL was in the range 18–29 AAs, while β-emulsifiers appeared to be best in the range 13–15 AAs [44]. This is in line with the PCL of the peptides identified here predicted as emulsifiers in the different conformations. Nevertheless, emulsifying activity of protein hydrolysates is a complex mechanism, as the mixture of peptides and their abundance is highly complex. This also means that it may not be straightforward to directly transfer knowledge from isolated peptide emulsifiers to hydrolysates, as the nature of the functionality is influenced by cooperative, cumulative, and inhibitory effects in the interplay between individual peptides [56]. This further implies that there may indeed also be peptides not found in the top 20 most abundant, which greatly influence the bulk properties. For instance, in the top 100 most abundant (Appendix A), there are several peptides with very high scores (>3), which may influence the FPH properties and hence interesting leads to investigate in further detail.

Although we previously argued that amphiphilicity scores cannot be directly correlated to foaming properties, there are several interesting peptide-level observations that may explain bulk foaming properties to some degree. In addition to the apparent PCL-dependency, Neut derived FPHs, which show the highest FC, contain several peptides with predicted α-helical amphiphilicity, and although they may not be the key peptides in the observed O/W emulsifying properties, they may very well be able to form a peptide film at the air/water interface. As secondary structure is a prerequisite for strong intermolecular interactions through self-assembly [57,58], the significantly larger proportion of peptides predicted to have a potentially amphiphilic, well-defined secondary structure at the interface may also explain why significantly higher FC is observed for single enzyme treatments compared to sequential hydrolysis. The lower stability of Neut derived FPH from HCM compared to MCF may therefore also be related to the lower EAI observed. Similarly, the higher FC and FS observed for Alc derived FPH from MCF compared to HCM may to a degree be ascribed to the presence of the high scoring β-emulsifier, as particularly β-strand peptides are associated with higher elasticity of interfacial layers/films [58,59].

### 2.4. Antioxidative Properties

#### 2.4.1. Bulk Antioxdiative Properties

The antioxidative activity of FPHs from both MCF and HCM, evaluated by DPPH radical scavenging activity (RSA) and chelating capacity of ferrous iron (MCA), is presented in Table 1. While RSA describes the ability to transfer an electron to a radical oxidant, MCA describes the ability to chelate metals and thereby potentially prevent the catalytic breakdown of lipid peroxides, which leads to the formation of reactive alkoxyl radicals and volatile oxidation products responsible for off-flavor formation [60]. DPPH RSA of all FPHs exhibited significantly different concentrations for obtaining 50% inhibition (IC50; *p* < 0.05). All FPHs from HCM showed higher DPPH RSA compared to FPHs from MCF, and the highest effectiveness was observed for HCM obtained by sequential application of Neut&Alc (IC50 = 2.9 mg/mL). In contrast, FPHs from MCF showed higher MCA compared to FPHs from HCM (*p* < 0.05). Previous studies of the antioxidative properties of cod frame FPHs are somewhat inconclusive in terms of the influence of DH [5,61], but, generally, antioxidant activity is higher for lower MW fractions. Those findings correspond well with previous studies indicating that antioxidant activity in hydrolysates is positively correlated with the increase of DH [16,62], and high antioxidant activity is usually seen for peptides spanning 5–16 AAs [8,50,63], i.e., the shorter peptides, the higher antioxidative activity. Foh et al. [16] attributed that higher functionality and bioactivity of Alc derived peptides to the peptides fraction with MW < 1 KDa, compared to FPH from Neut and Flavourzyme that was characterized by a high percentage of peptides with MW ranging from 8 kDa to 15 kDa. Nevertheless, our data do not suggest a general correlation between increased hydrolysis and increased antioxidant activity, although several studies have found antioxidative peptides to be rather short and usually in the range 5–16 AAs [8,50,63,64]. However, the highest DPPH RSA was found for the FPHs obtained by sequential hydrolysis (i.e., highest DH and lowest PCL) for both MCF and HCM compared to other treatments on the same substrate [65]. There does not appear to be any apparent correlation with the mean peptide charge, Pz_avg_ (Table 2) for neither RSA nor MCA.

As discussed for emulsifying activity, considering hydrolysis and PCL alone as a determining factor for antioxidant activity is naive. Studies have shown that changes in concentration and composition of the free amino acids [66] and small peptides [67] influence the antioxidative activity of protein hydrolysates [68]. Furthermore, the presence of specific amino acids and their distribution and special orientation contribute to the overall antioxidative activity. This covers inclusion of aromatic AAs such as Tyr, Trp, and Phe [47,69,70], hydrophobic AAs including Val, Leu, and Ala [47,69,70,71,72] sulphur-containing AAs, i.e., Met and Cys [47,69,70], acidic AAs, i.e., Glu and Asp [71,73], alkaline AAs, i.e., His, Lys and Arg [47,70,74,75], as well as Pro [8], which all may dramatically influence the antioxidative activity of peptides. Furthermore, high content of N-terminal Leu, Ala and Val has also been reported to positively affect antioxidant activity [8,76]. This means that it is highly problematic to make any generalized assumptions regarding AA composition in antioxidative peptides, as this is of a highly complex nature.

#### 2.4.2. Interconnection of Bulk Antioxidative Properties, Peptide Abundance, and Predicted Antioxidative Activity

In order to obtain a better understanding of the underlying principles for peptide antioxidative activity, computational approaches are needed. For this specific application, AnOxPePred, a neural network algorithm for prediction of antioxidant activity, has been developed using verified antioxidative peptides as input for feature and pattern recognition [77]. Predicted free radical scavenging (FRS) and metal chelating activity (MCA) for the 20 most abundant peptides from each FPH are presented in Table 4, while the scores for the 100 most abundant peptides from each FPH are presented in Appendix A. The lowest in vitro DPPH RSA (i.e., highest IC50) was for both substrates determined in the Alc derived FPH. Surprisingly, the Alc derived FPH for both substrates show the highest number (and cumulative abundance) of peptides predicted to display RSA (i.e., scores > 0.43) [77]. Furthermore, the highest scoring peptide of all, LLPVLYPPVVEE (FRS = 0.64), was found in both Alc derived FPHs. Nevertheless, when comparing the abundance of the peptide, it is significantly more abundant in the FPH from HCM (1.09% vs. 0.65%, i.e., 1.7-fold). This is also observed for another high scoring (FRS = 0.56) peptide VAPEEHPTLL (1.87% vs. 1.12%, i.e., 1.7-fold), which may explain the higher DPPH RSA observed for the Alc derived FPH from HCM compared to the FPH from MCF. The peptide SGSAGKDGMSGLPGPSGPSGPRGR was also predicted to be a FRS with a high score (FRS = 0.55) but was present in equivalent abundance in both FPHs (0.67%). A number of other peptides with predicted RSA above the score threshold (FRS = 0.43–0.51) was also found in both FPHs, but their relative abundance varied between samples, thereby making it difficult to assess their influence on the bulk RSA.

For both MCF and HCM, the FPH obtained by sequential hydrolysis was observed to display the highest RSA, when comparing between treatments, with the FPH from HCM showing the highest activity. This is surprising, as the highest scoring peptide from Neut&Alc FPHs, i.e., GVDNPGHPF, is two-fold more abundant in the MCF FPH (2.0% vs. 1.0%). Consequently, this peptide cannot explain the difference observed. In contrast, two other peptides with FRS scores above the threshold are more abundant in the HCM FPH, VAPEEHPTL (FRS = 0.51–1.91% vs 1.59%), VLYPPVVEE (FRS = 0.52–1.75% vs. 1.53%), and may be involved in the higher activity observed. Furthermore, the tetrapeptide FLPM (FRS = 0.46) is not observed in the top 20 abundant peptides of the MCF FPH (RI = 0.08%, Appendix A) but is highly abundant (1.1%) in the HCM FPH, which also makes this a highly interesting peptide, in terms of describing the observed differences. FLPM is also observed in the Neut derived FPH from HCM but with lower abundance (0.66%). This may also be a part of the lower DPPH RSA observed for this FPH compared to the sequential FPH from HCM, but also the higher DPPH RSA compared to the Neut derived FPH from MCF, where the peptide is not observed in significant abundance (0.04%, Appendix A).

In terms of MCA, only one peptide with a score over the threshold (MCA > 0.3) was observed in the top 20 abundant peptides across all samples. The peptide, FLPM (MCA = 0.31), was also predicted to be a potential FRS as discussed above. Nevertheless, the peptide was identified with the highest abundance in the sequential hydrolysate from HCM, which was in fact the FPH with the lowest MCA of all six FPHs. Consequently, there are likely other significantly different factors describing the observed differences than predicted by AnOxPePred. For metal chelation, secondary and tertiary structure may be a very important factor, as for instance metalloenzymes, which coordinate cationic species, usually have highly complex and well-defined coordination sites, which are highly influenced by structural changes [78,79].

When comparing between substrates, the FPH obtained from HCM did in all cases display the highest RSA, whereas the FPHs from MCF displayed the highest MCA, indicating the substrate used for hydrolysis is of importance. For instance, FPHs from HCM contain a significantly higher amount of calcium [2]. As calcium is a divalent cation similar to Fe^2+^, a higher content may explain why we observe a generally lower MCA for FPHs obtained from HCM. Bones also contain a significant amount of free radicals or reactive oxygen species (ROS) produced by osteoclasts [80,81] as part of bone metabolism and signaling. As MCF includes cut bones during hydrolysis, they may be the source of ROS, which in turn will impair the RSA of the hydrolysates, as the FPH already contains ROS and/or FRS peptides may have already donated an electron to endogenous ROS. This could be the underlying reason of FPHs from MCF generally displaying lower RSA. Consequently, it could be argued that, due to these factors, FPHs from the different substrates may be hard to compare directly in terms of in vitro antioxidant activity.

When predicting peptide functionality and bioactivity using computational methods, it is of great importance to consider and differentiate between background and function of an algorithm. EmulsiPred calculates an amphiphilicity vector based on the AA distribution in a given conformation. This means that it models the peptide and then determines a physio-chemical property, which is independent of a training dataset and can be regarded as a global, sequence-dependent constant attribute. In contrast, AnOxPePred uses a neural network that has been trained on experimental data, and consequently relies heavily on the size and the quality of the dataset. As limited, high-quality data for this, in particularly true negatives, exist, the model performance may be suboptimal [77]. This is especially the case for MCA prediction, where the training dataset is very limited. These observations may also explain why prediction of emulsifying activity appears to describe the unexpected trends (in terms of DH-dependency) quite well, while the predictions for both RSA and MCA appear to be somewhat lacking in describing the observed differences. Limitations in the prediction model may also explain why high scoring peptides appear to adhere well to the overall characteristics for antioxidant peptides outlined above. Furthermore, as mentioned for emulsifying peptides, there may very well be peptides not included in the top 20 abundant that influence the bulk properties. Indeed, there are several peptides in the top 100 (Appendix A) which have very high FRS (>0.6) as well as peptides with significant MCA (>0.3), which may contribute to the observed bulk properties. Nevertheless, this study has identified a large number of promising peptides with predicted functionality, which allow for further investigation of peptide-level functionality.

## 3. Materials and Methods

Cod frames were received from Espersen Company, (Espersen A/S, LT 94102, Klaipeda, Lithuania) in a batch of 7 kg (September 2017) in frozen condition stored at −40 °C until hydrolysis. Alcalase^®^ 2.4 L FG (Alc; declared activity of 2.4 AU/g) and Neutrase^®^ 0.8 L (Neut; declared activity 0.8 AU/g) was provided by Novozymes Company (Novozymes A/S, 2880, Bagsværd, Denmark). Sodium caseinate (MIPRODAN 30) was provided by Arla Company (Arla Foods Ingredients amba, Viby J, Denmark). All chemical reagents used for experiments were of analytical grade.

### 3.1. Enzymatic Hydrolysis Procedure

The enzymatic hydrolysis carried out as described in Jafarpour et al. [2]. Briefly, cod frames were cut into smaller parts and then divided in two batches; minced cod frame (MCF) and heated cod meat (HCM) were boiled for 20 min at 95 °C, and the bones were subsequently removed. Alc and Neut enzymes were applied either individually (3 h) or sequentially (2 × 3 h) and proteolysis was performed at 50 °C, pH 7.4, using an enzyme/substrate ratio of 1.5%.

#### Degree of Hydrolysis (DH)

DH was estimated from α-amino nitrogen following Nielsen et al. [28];
(1)$$DH\%=\frac{\alpha AN_{hyd}}{\alpha AN_{tot}}\times 100\%$$
where $\alpha AN_{hyd}$ is the α-amino-N content of the hydrolysates (mM/g) and $\alpha AN_{tot}$ is the α amino-N content of the fully hydrolyzed substrate (mM/g). $\alpha AN_{tot}$ was calculated based on the HPLC-MS total amino acid analysis and protein content of the untreated substrate previously reported [2]. The $\alpha AN_{hyd}$ was determined using the PractiChrom^TM^ PFAN-25 free amino nitrogen assay kit (BioAssay Systems, Hayward, CA, USA) by application of Picoexplorer™ apparatus (model PAS-110, USHIO, Cypress, CA, USA). Accordingly, the ninhydrin-based reaction was performed by adding of 150 µL of reagent in 5 µL of dissolved FPH, and incubating for 10 min at 100 °C. Thereafter, the reaction mixture was cooled to room temperature by leaving on bench for 15 min. Absorbance was measured at 530 nm and standard curve was obtained using glycine (BioAssay Systems, Hayward, CA, USA) as free α-amino group. Based on the determined DH for each FPH, the average peptide chain length (PCL) for the hydrolysate was calculated according to [22], as:(2)$$PCL=\;\frac{1}{DH\;\left(\%\right)}$$

### 3.2. Emulsifying Properties

The emulsifying activity index (EAI) and the emulsion stability index (ESI) were determined using the method described by [17,82] with slight modification. Accordingly, 15 mL portions of 2 mg/mL protein solution of each hydrolysate sample were mixed with 5 mL rapeseed oil using an ultraturax homogenizer (IKA, Staufen, Germany) at a speed of 9500 rpm for 60 s without pH adjustment. Afterward, aliquots of 50 μL were pipetted from the bottom of the container at 0 and 10 min after homogenization and then mixed with 5 mL of 0.1% sodium dodecyl sulfate (SDS) (Sigma-Aldrich, St. Louis, MO, USA) solution in order to measure the absorbance of the diluted solution at 500 nm using a spectrophotometer (Shimadzu, UV-1280, Kyoto, Japan). EAI was calculated as:(3)$$EAI\left(\frac{m^{2}}{g}\right)=\frac{2\times 2.303\times A_{0}\times D}{\emptyset \times c\times 10000}$$
where *A*_0_ is the absorbance immediately following homogenization, *D* is the dilution factor, ∅ is the oil volume fraction, and *c* is protein concentration (g/mL).

ESI was calculated as:(4)$$ESI\left(min\right)=\left(\frac{A_{0}}{A_{0}-A_{10}}\right)\times 10\;min$$
where *A*_0_ is the absorbance immediately following homogenization and *A*_10_ is the absorbance at 10 min. Sodium caseinate (2 mg/mL) protein solution was used as control. Measurements were performed in triplicates.

### 3.3. Foaming Properties

Foaming capacity (FC) and foaming stability (FS) were determined according to the method of [83]. Accordingly, 25 mg of each protein hydrolysate sample were dissolved in 25 mL of distilled water solution (0.1 % *w*/*v*) and mixed by a magnet stirrer for 3 min and then homogenized at 9500 rpm for 120 s using an Ultraturrax (IKA, Staufen, Germany), and finally poured into a 200 mL graduated cylinder. FC was determined by recording the foam volume immediately after homogenization and calculated as:(5)$$FC\left(\%\right)=\frac{V_{foam}^{0}}{V_{init}}\times 100\%$$
where $V_{foam}^{0}$ is the foam volume immediately after homogenization and $V_{init}$ is the initial sample volume before homogenization.

FS is calculated after a 30 min resting period as:(6)$$FS\left(\%\right)=\frac{V_{foam}^{30}}{V_{foam}^{0}}\times 100\%$$
where $V_{foam}^{30}$ is the foam volume after 30 min and $V_{foam}^{0}$ is the foam volume immediately after homogenization.

### 3.4. In Vitro Antioxidant Properties

#### 3.4.1. Diphenyl-1-Picryhhydrazyl (DPPH) Radical Scavenging Activity (RSA)

Solutions of FPH at different concentrations were added to an equal volume of 0.1mM DPPH (Sigma-Aldrich, St. Louis, MO, USA) in 96% EtOH (Sigma-Aldrich, St. Louis, MO, USA) in a microplate (Eon 2, BioTek, Winooski, VT, USA). A blank of each sample (sample + EtOH) was included as a negative control. Three blind samples (EtOH + DPPH) were also prepared on the plate. The plate was incubated for 30 min at room temperature in darkness before measuring the absorbance at 517 nm using a microplate reader (Eon 2, BioTek, Winooski, VT, USA). Scavenging activity was calculated as inhibition percentage (I_RS_) using mean absorbance as:(7)$$I_{RS}\left(\%\right)=\left(1-\frac{A_{s}-A_{0}}{A_{b}}\right)\times 100\%$$
where *A_s_* is the absorbance of DPPH after reaction with antioxidant, *A*_0_ is the absorbance of antioxidant and EtOH negative control), and *A_b_* is the absorbance of EtOH and DPPH (blind).

Based on the inhibition percentages for the different concentration for each FPHs, RSA was determined as the IC_50_ based on linear regression. Each sample concentration was measured in triplicate.

#### 3.4.2. Metal Chelating Activity (MCA)

In addition, 100 µL of solution of each FPH at different concentrations was added to a microplate (Eon 2, BioTek, Winooski, VT, USA). Furthermore, 0.5 mM EDTA (Sigma-Aldrich, St. Louis, MO, USA) was used as positive control. To start the reaction, 20 µL of 0.5 mM ferrous chloride (Sigma-Aldrich, St. Louis, MO, USA) was added to the plate and mixed in a plate mixer for 3 min at 300–600 rpm. Subsequently, 20 µL of 2.5 mM ferrozine (Sigma-Aldrich, St. Louis, MO, USA) was added. Blanks consisted of only iron chloride and Ferrozin. The plate was mixed and left for 10 min in dark at room temperature. The absorbance at 562 nm was measured using a microtiterplate-reader (Eon 2, BioTek, Winooski, VT, USA). Ferrous chelating activity (FCA) was calculated as inhibition percentage (*I_FCA_*) using mean absorbance as:(8)$$I_{FCA}\left(\%\right)=\left(\frac{A_{\mathrm{blank}}-\left(A_{\mathrm{sample}}-A_{\mathrm{blind}}\right)}{A_{\mathrm{blank}}}\right)\times 100\%$$
where *A_blank_* is the absorbance of blank, *A_sample_* is the absorbance of sample, and *A_blind_* is the absorbance of blind (only FPH).

Based on the inhibition percentages for the different concentration for each FPHs, the MCA was determined as the EC_50_ based on linear regression. Each sample concentration was measured in triplicate.

### 3.5. 1D SDS-PAGE Analysis

Freeze-dried FPH samples were solubilized with 2% SDS (AppliChem GmbH, Darmstadt, Germany) in 200 mM ammonium bicarbonate (Sigma-Aldrich Chemie GmbH, Steinheim, Germant) (pH 9.5) to a final protein/peptide concentration of 2 mg/mL based on determined protein content by Dumas, as previously described [2]. Samples were vortexed for 2 min and sonicated for 30 min. Subsequently, samples were centrifuged at 4000× *g* for 15 min to precipitate insoluble proteins/peptides and solids using a 5810 R centrifuge (Eppendorf, Hamburg, Germany). SDS-PAGE analysis was performed using precast Criterion 16.5% Tris-Tricine gels (Bio-Rad, Hercules, CA, USA) under reducing conditions according to manufacturer guidelines by loading 20 µg protein/peptide in Tricine sample buffer (Bio-Rad) containing dithiothreitol (Roche Diagnostics GmbH, Mannheim, Germany)) to a final concentration of 50 mM (denatured at 95 °C for 5 min). As molecular weight (MW) marker, Unstained Polypeptide SDS-PAGE Standard P/N 1610326 (Bio-Rad) was used. To alleviate low signal intensity and smearing, an identical gel was prepared but loading 10-fold more protein/peptide in each well. For full range visualization, SDS-PAGE analysis was also performed on precast 4-20% gradient gels (GenScript, Piscataway, NJ, USA) in a Tris-MOPS buffered system under the same general conditions as above using both 20 µg and 200 µg protein/peptide per well. As the MW marker, PIERCE Unstained Protein MW Marker P/N 26610 (ThermoFisher Scientific, Rockford, IL, USA) was used. Protein/peptide visualization was achieved with Coomassie Brilliant Blue G250 (Sigma-Aldrich Chemie GmbH) staining using a ChemDoc MP Imaging System (Bio-Rad).

### 3.6. Reference Proteome Construction

The reference sequence database for peptide identification was derived from two existing assemblies: GadMor_May2010 [84] and ASM90030256v1 [85], as previously described [2]. The proteome contained 74913 protein entries with an average length of 325.7 amino acids (AAs), calculated using the Sequence Length FASTA Tool [86].

### 3.7. LC-MS/MS and Proteomics Data Analysis

FPH samples were analyzed as previously reported [2]. In short, hydrolysates were reduced and alkylated in-solution before desalting using C-18 StageTips. Analysis was performed using an EASY-nLC system (Thermo Scientific, Bremen, Germany) online coupled to a Q Exactive HF mass spectrometer (Thermo Scientific) following separation on a RP Acclaim Pepmap RSLC analytical column (C18, 100 Å, 75 μm. × 50 cm, nanoViper fittings (Thermo Scientific).

Data analysis was performed as previously described using MaxQuant 1.6.0.16 [36,37] with the constructed reference proteome. The analysis was performed using unspecific digestion and a defined peptide length of 3 to 65 AAs. A false discovery rate of 1% was applied on both the peptide and protein level.

#### 3.7.1. Post-Processing of Proteomics Data

Building on the principles of intensity-weighed peptide abundance estimation previously outlined [2], we defined a number of parameters for sample characterization and comparison. These parameters relate to the average peptide chain length (PCL_avg_), average peptide molecular weight (PMW_avg_), and average peptide charge (Pz_avg_) at physiological conditions (pH 7) determined as weighted means according to:(9)$$PCL_{avg}=\frac{\sum_{p=1}^{n}PCL_{p}*I_{p}}{\sum_{p=1}^{n}I_{p}}$$
(10)$$PMW_{avg}=\frac{\sum_{p=1}^{n}PMW_{p}*I_{p}}{\sum_{p=1}^{n}I_{p}}$$
(11)$$Pz_{avg}=\frac{\sum_{p=1}^{n}Pz_{p}*I_{p}}{\sum_{p=1}^{n}I_{p}}$$
where *PCL_p_*, *PMW_p_*, and *Pz_p_* is the length, MW, and charge at pH 7 for a given peptide *p* of all *n* peptides identified in a sample and *I_p_* is the intensity of peptide *p* (the weight). For comparison, unweighted averages were also calculated and plotted for each parameter. Peptide charge was calculated using the principles applied in the peptide property calculator from INNOVAGEN (Innovagen AB, Lund, Sweden), which is based on pKa values of individual AAs [87]. At neutral pH, the following AAs contribute to the peptide net charge (contribution given in parenthesis): Lys (+1), Arg (+1), His (+0.1), Asp (−1), Glu (−1), and Cys (−0.07).

To evaluate variance of length, MW, and charge in the population distributions, the standard deviation of weighted means was calculated as:(12)$$SD=\sqrt{\frac{\sum_{p=1}^{n}I_{p}{\left(x_{p}-x_{avg}\right)}^{2}}{\frac{\left(M-1\right)}{M}\sum_{p=1}^{n}I_{p}}}$$
where *I_p_* is the intensity of peptide *p*, *x_p_* is the parameter observation for peptide *p* (e.g., PCL_p_), *x_avg_* is the weighted population mean for the parameter (e.g., PCL_avg_), and *M* is the number of non-zero weights.

For simplicity in visualization, data were binned for histogram depiction. PCL was binned with a bin size of 5 AAs (i.e., 3–7 AAs were binned and labeled as 5 AAs, etc.). PMW was initially binned around each 100 Da (i.e., all peptides with 150 < PMW < 249.9999 were all labelled as PMW = 200 Da, and subsequently binned again around each 500 Da (i.e., 150 < PMW < 749.9999 was labelled PMW = 500, 750 < PMW < 1249.9999 was labelled PMW = 1000 Da etc.). Due to low occupancy in the upper bins, all peptides with PMW > 4250 Da are represented in the PMW = 4500 Da bin. Pz was initially binned around each integer charge (i.e., all peptides with −0.5 < Pz < 0.49 were labelled as Pz = 0, etc.) and, subsequently, the most extreme charge states were binned (i.e., Pz < −6.51 were all labeled as Pz = −7 and Pz > 2.5 were all labelled as Pz = 3). Histograms of PCL, PMW, and Pz distributions were plotted in OriginPro 8.5.0 SR1 (OriginLab Corporation, Northampton, MA, USA) and figures assembled in their final form using INKSCAPE version 0.92.3 [88].

#### 3.7.2. Prediction of Emulsification Activity

The computational prediction of emulsifying properties was performed using the EmulsiPred algorithm described by [44], an improved version of an existing method [89], which is freely available [90]. To give a short overview, three different emulsifier scores, namely *s_α_*, *s_β_*, and *s_γ_*, are computed based on the sequence of peptides, based on the potential ability of the peptide to assume an α-helix, β-strand, or undefined conformation at the oil–water interface, respectively. The only difference with the original algorithm is that the score for a generic peptide *p* is normalized based on the following formula:(13)$$z_{i}=\frac{s_{i}-µ\left(s_{i}\right)}{\sigma \left(s_{i}\right)}$$
where *s_i_* is one of the scores *s_α_*, *s_β_*, or *s_γ_*, calculated for *p*, and µ($s_{i}$) and $\sigma \left(s_{i}\right)$ are the mean and the standard deviation of the same score, calculated from a population of 40,000 peptides of the same length as *p*. The emulsifying scores are therefore normalized around 0, with positive normalized scores for emulsifying scores higher than the average and negatives normalized scores for emulsifying scores lower than the average. This normalization procedure assures that scores obtained from peptides of different length have a similar distribution and can be more easily compared. Emulsifying activity was predicted for the 100 most abundant peptides (by peptide MS1 intensity) for each FPH sample fulfilling a requirement of PCL > 6 (requirement to form any well-defined secondary structure at the interphase).

#### 3.7.3. Prediction of Antioxidant Activity

For the computational predictions of antioxidant peptides, the AnOxPePred webserver (available at http://services.bioinformatics.dtu.dk/service.php?AnOxPePred-1.0) was used [77]. This predictor is based on a machine learning approach trained on a free radical scavenger (FRS) and metal chelating data (MCA) set, with its predictions being from 0 (not antioxidant) to 1 (antioxidant). Antioxidant activity was predicted for the 100 most abundant peptides (by peptide MS1 intensity) for each FPH sample fulfilling a requirement of PCL ≤ 30.

### 3.8. Statistical Analysis

The current study was performed in a complete randomized block design test and obtained data were analyzed in one-way ANOVA using STATGRAPHICS (version 18.1.06 The Planis, VA, USA). Multiple comparison among means was calculated in Tukey HSD as a post-hoc test, while setting the confidence level at 95%.

## 4. Conclusions

Short time enzymatic hydrolysis of a cod frame by application of Neut and Alc, individually or sequentially, released peptides from minced cod frame and heated cod meat, showing promising interfacial activity and antioxidative properties. Using a novel proteomic approach, we were able to determine a number of weighted bulk attributes of the hydrolysates as well as an in-depth peptide-level characterization. Combined with bioinformatic prediction of peptide functionality in a hitherto unseen way, we were able to identify a number of highly specific peptides, which are both highly abundant and predicted with high probability as being key in the observed activity. We observed a general and surprising trend that increased hydrolysis did in fact increase emulsifying activity. Nevertheless, this observation correlated with two highly abundant peptides (IIAPPERKYS and GADPEDVIVA) with a high potential as γ-emulsifiers (containing both a hydrophobic and a hydrophilic region). Furthermore, we identified a number of peptides that are highly likely to have emulsifying activity in a well-defined, secondary structure, in particular one β-strand emulsifier (DIDIRKDLYAN), which are all promising leads. We also identified a number of peptides, predicted to have antioxidant properties in vitro, although some substrate-specific differences were observed, which might be related to the presence of residual bone in the minced cod frame. Our results demonstrate that using both proteomics analysis and bioinformatic prediction is of great value when characterizing protein hydrolysates for use in food and feed.

## Figures and Tables

**Figure 1 marinedrugs-18-00599-f001:**
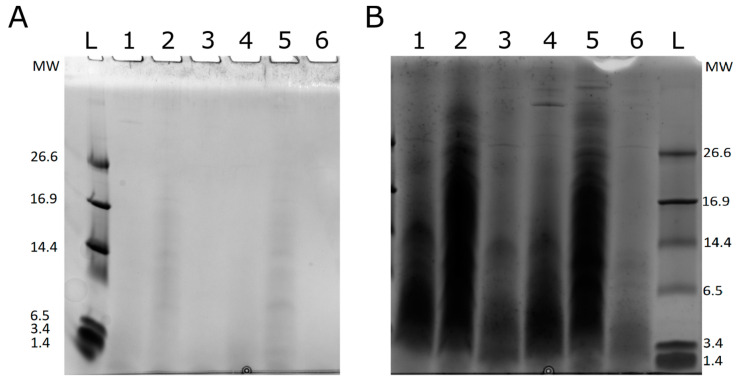
SDS-PAGE of FPH samples on 16.5% Tris-Tricine gels. (**A**) 20 μg protein/peptide [2]; (**B**) 200 μg protein/peptide [2]. L (MW marker), 1 (MCF-Alc), 2 (MCF-Neut), 3 (MCF-Neut&Alc), 4 (HCM-Alc), 5 (HCM-Neut) and 6 (HCM-Neut&Alc).

**Figure 2 marinedrugs-18-00599-f002:**
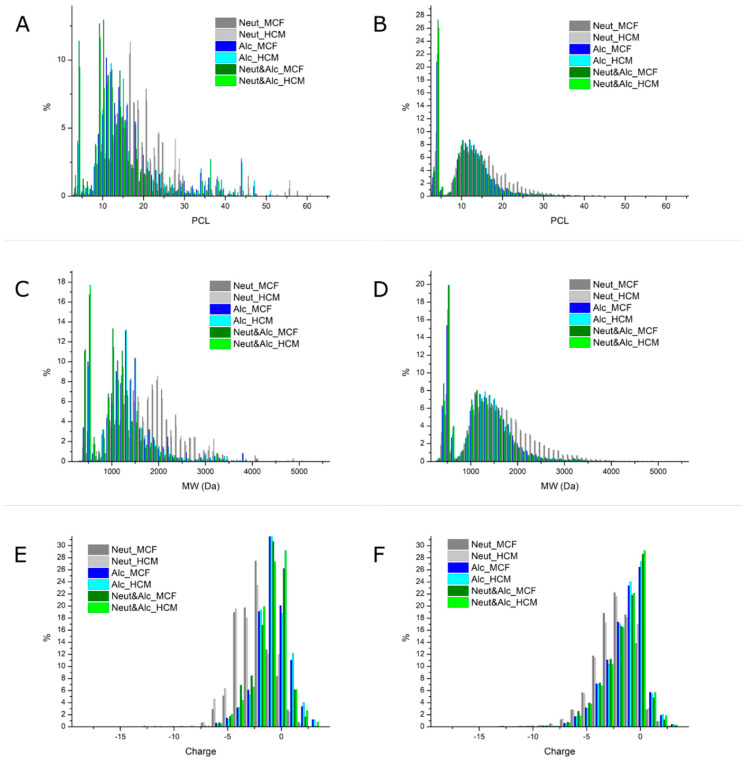
Histograms of unbinned, relative distributions for peptide chain length (**A + D**), peptide molecular weight (**B + E**), and peptide charge at pH = 7 (**C + F**) for fish protein hydrolysates obtained from minced cod frame (MCF) and heated cod meat (HCM) by enzymatic hydrolysis with neutrase (Neut) in greys, alcalase (Alc) in blues, or sequential hydrolysis with neutrase and alcalase (Neut&Alc) in greens. Distributions are shown for intensity-weighted data (**A**–**C**) and unweighted data (**D**–**F**).

**Table 1 marinedrugs-18-00599-t001:** Bulk characterization of functional properties in fish protein hydrolysates (FPH). Degree of hydrolysis (DH), emulsification activity index (EAI), emulsification stability index (ESI), foaming capacity (FC), foaming stability (FS), DPPH radical scavenging activity, and metal chelating activity of FPH from minced cod frame (MCF) and heated cod meat (HCM) by application of Alcalase and Neutrase separately or sequentially. For comparison, sodium caseinate (SC) was used as control. N/A: Not applicable (SC was used as is).

Substrate	Treatment	DH (%) *	EAI (m^2^/g) *	ESI (min) *	FC (%)	FS (%)	DPPH RSA (IC50 mg/mL) *	Fe^2+^ MCA (IC50 mg/mL) *
MCF	Neutrase	27.0 ± 0.12 ^d^	80.75 ± 4.32 ^ab^	33.39 ± 0.56 ^cd^	40	88	4.36 ± 0.08 ^b^	0.53 ± 0.013 ^c^
	Alcalase	36.1 ± 0.03 ^b^	88.18 ± 7.94 ^a^	29.38 ± 3.52 ^d^	30	87	4.97 ± 0.18 ^a^	0.77 ± 0.03 ^bc^
	Neutrase & Alcalase	39.2 ± 0.04 ^a^	87.54 ± 4.46 ^a^	36.10 ± 5.12 ^c^	4	100	4.31 ± 0.06 ^b^	0.77 ± 0.03 ^bc^
HCM	Neutrase	18.7 ± 0.09 ^e^	66.41 ± 0.97 ^c^	46.82 ± 4.75 ^b^	54	37	3.36 ± 0.18 ^c^	0.69 ± 0.02 ^c^
	Alcalase	30.2 ± 0.05 ^c^	68.35 ± 0.72 ^c^	65.15 ± 2.03 ^a^	23	78	3.58 ± 0.21 ^c^	1.01 ± 0.31 ^b^
	Neutrase & Alcalase	30.2 ± 0.04 ^c^	76.48 ± 3.99 ^b^	45.70 ± 1.94 ^b^	6	100	2.93 ± 0.05 ^d^	1.65 ± 0.07 ^a^
Control (SC)	-	N/A	64.48 ± 3.33 ^c^	15.86 ± 1.15 ^e^	N/A	N/A	N/A	N/A

Different small superscript letters in each column indicate the significant differences among means at 95 confidence level (α = 0.05). * Mean ± SD. All data are based on three replicates.

**Table 2 marinedrugs-18-00599-t002:** Number of identified peptides, average peptide chain length (PCL_avg_), average peptide molecular weight (PMW_avg_), and average peptide charge (Pz_avg_) as determined by LC-MS/MS analysis. Mean peptide attributes are expressed as both weighted (by peptide MS1 intensity) and unweighted. PCL calculated as inverse degree of hydrolysis (PCL_DH_) according to Adler-Nissen (1986) is listed for comparison [28].

	Treatments		Weighted			Unweighted			DH-based
Substrate	Treatment	Peptide IDs	PCL_avg_ (AAs) ^1^	PMW_avg_ (Da) ^1^	Pz_avg_ ^1^	PCL_avg_ (AAs) ^1^	PMW_avg_ (Da) ^1^	Pz_avg_ ^1^	PCL_DH_ (AAs) *
MCF	Neutrase	6268	17.4 ± 6.9	1921 ± 674	−2.5 ± 1.7	15.3 ± 7.1	1686 ± 726	−2.2 ± 1.9	3.7 ± 0.0 ^b^
	Alcalase	4414	13.0 ± 7.3	1343 ± 635	−0.95 ± 1.6	11.7 ± 6.5	1268 ± 636	−1.5 ± 2.0	2.8 ± 0.0 ^d^
	Neutrase & Alcalase	3702	9.36 ± 4.8	999 ± 463	−1.3 ± 1.6	10.3 ± 6.1	1128 ± 614	−1.6 ± 2.0	2.6 ± 0.0 ^e^
HCM	Neutrase	5871	16.7 ± 7.3	1853 ± 722	−2.5 ± 1.8	14.8 ± 7.4	1635 ± 755	−2.2 ± 2.0	5.4 ± 0.0 ^a^
	Alcalase	4020	13.1 ± 7.7	1334 ± 647	−0.88 ± 1.6	11.3 ± 6.5	1228 ± 622	−1.4 ± 1.9	3.3 ± 0.0 ^c^
	Neutrase & Alcalase	4361	10.2 ± 6.6	1082 ± 591	−1.1 ± 1.6	10.6 ± 6.3	1157 ± 611	−1.5 ± 2.0	3.3 ± 0.0 ^c^

Different small superscript letters in each column indicate the significant differences among means at 95 confidence level (α = 0.05). ^1^ No significant difference between means at a 95% confidence level (α = 0.05). * Mean ± SD. All data are based on three replicates.

**Table 3 marinedrugs-18-00599-t003:** Sequences, relative intensity (RI), and predicted emulsifying activity (score) in α-helical, β-strand, and γ (axial amphiphilicity) conformation for the 20 peptides with highest relative intensity (PCL > 6) from enzymatically derived FPH from minced cod frame (MCF) and heated cod meat (HCM). Emulsifier prediction scores > 2 (topmost 2.5%) are highlighted in bold.

**MCF**														
**Neut.**	**RI (%)**	**Emulsifying Scores**			**Alc**	**RI (%)**	**Emulsifying Scores**			**Neut&Alc**	**RI (%)**	**Emulsifying Scores**		
		**α**	**β**	**γ**			**α**	**β**	**γ**			**α**	**β**	**γ**
LQGEVEDLMVDVERANG	1.94	1.38	−0.45	0.55	GFAGDDAPRAVFPS	2.30	0.29	−0.86	0.71	IIAPPERKYS	3.49	−0.59	−0.62	**3.43**
LEQQVDDLEGSLEQEKK	1.92	**2.18**	−0.87	1.26	RVAPEEHPTLL	2.13	0.36	−0.09	1.79	GVDNPGHPF	2.00	1.13	−0.49	0.08
IITNWDDMEK	1.78	0.75	−1.28	**2.95**	AGDDAPRAVFPS	2.09	0.97	−1.33	1.23	VAPEEHPTL	1.59	−0.18	−0.07	1.04
VQHELEEAEERADIAETQVNK	1.39	**2.21**	−0.41	0.64	AGPAGPSGPRGPAGIA	1.56	0.16	−0.17	0.47	VLYPPVVEE	1.53	−0.43	−1.16	0.55
LTKLEEAEKAADESERGMK	1.17	1.53	−0.55	1.15	KSYELPDGQVITIG	1.33	−0.78	−0.79	**2.64**	GADPEDVIVA	1.43	−1.29	−1.00	**3.10**
LEDQLSEIKAKSDENARQ	1.00	1.54	0.10	1.33	GAAGPAGPSGPRGPAGIA	1.30	0.16	−0.51	0.23	VIDQDKSGFIE	1.35	0.37	−0.43	0.88
LEKSYELPDGQVIT	0.92	−0.55	−0.43	0.82	DIDIRKDLYAN	1.16	−1.24	**3.65**	0.83	AGDDAPRAVFPS	1.20	0.97	−1.33	1.23
VAPEEHPTL	0.90	−0.18	−0.07	1.04	VAPEEHPTLL	1.12	−0.53	−0.81	1.14	KSYELPDGQ	1.19	−1.09	−0.94	−0.55
IIDQNRDGIISKDDLRD	0.86	−0.51	−0.56	0.70	AGPSGPRGPAGIA	1.08	−0.23	−0.12	1.26	GERGEQGPGGPGGF	1.08	−0.79	−0.73	1.36
LDDLQAEEDKVNT	0.81	0.55	−1.02	1.17	NWDDMEKIWHH	1.06	0.75	−0.27	1.02	IIDQNRDGIIS	0.83	0.68	−0.86	1.67
LEKTIDDLEDELYAQK	0.78	1.51	−0.63	0.12	GQKDSYVGDEAQSKRGILTL	0.92	−0.71	−0.30	1.27	AGPAGPSGPRGPAG	0.81	−0.56	−0.48	0.21
LKGTEDELDKYSEALKDAQEKLE	0.77	**2.96**	0.22	0.47	LRVAPEEHPTL	0.88	0.64	1.15	1.24	GLPGPSGPSGPRGR	0.78	−0.81	−1.11	0.80
LTEEMASQDESVAK	0.76	1.01	−0.17	0.11	RGDSGPAGPPGEQGML	0.84	−1.16	−0.62	0.13	RGEQGPGGPGGF	0.76	−1.42	−0.18	1.04
LKGADPEDVIVAA	0.76	−1.53	−0.19	**2.78**	ELPDGQVITIG	0.82	−0.99	0.01	**2.07**	GFAGDDAPRA	0.69	0.45	−0.67	1.06
LADWKQKYEEGQAELEGSLKEARS	0.71	0.70	−0.08	0.43	SGSAGKDGMSGLPGPSGPSGPRGR	0.67	−0.23	−1.12	0.32	AINDPFIDL	0.67	0.83	−0.47	0.47
SKYETDAIQRTEELEESKKK	0.69	0.71	−0.69	1.57	LLPVLYPPVVEE	0.67	0.24	−1.23	1.33	KAGDSDGDGAIGVD	0.59	−0.30	0.65	1.81
LKAGDSDGDGAIGVDEWAV	0.69	0.26	−0.72	1.09	SGPGGPTGPSGM	0.64	−1.24	−0.74	−0.49	GFAGDDAPRAVFPS	0.58	0.29	−0.86	0.71
LTDAETKAF	0.65	0.43	−0.14	−0.44	KSYELPDGQVITIGNE	0.62	−0.17	−0.82	1.35	LEDQLSELK	0.54	1.36	−1.20	−0.22
LKAGDSDGDGAIGVDEWAVLVKA	0.61	−0.98	−1.16	1.53	GPAGPSGPRGPAGIA	0.59	−0.10	−0.37	0.91	KILDPEAT	0.52	0.59	−0.71	0.96
VDDIIQTGVDNPGHPFIMT	0.59	0.89	0.34	0.07	FAGDDAPRAVFPS	0.58	0.35	−0.89	0.70	GKDGMSGLPGPSGPSGPRGR	0.52	−0.08	−0.72	0.47
**HCM**														
**Neut.**	**RI (%)**	**α**	**β**	**γ**	**Alc**	**RI (%)**	**α**	**β**	**γ**	**Neut&Alc**	**RI (%)**	**α**	**β**	**γ**
LEQQVDDLEGSLEQEKK	2.38	**2.18**	−0.87	1.26	AGDDAPRAVFPS	3.13	0.97	−1.33	1.23	VAPEEHPTL	1.91	−0.18	−0.07	1.04
LQGEVEDLMVDVERANG	2.28	1.38	−0.45	0.55	GPAGPSGPRGPAGIA	2.64	−0.10	−0.37	0.91	VLYPPVVEE	1.75	−0.43	−1.16	0.55
IITNWDDMEK	2.23	0.75	−1.28	**2.95**	AGPAGPSGPRGPAGIA	1.90	0.16	−0.17	0.47	IIDQNRDGIIS	1.45	0.68	−0.86	1.67
VQHELEEAEERADIAETQVNK	1.46	**2.21**	−0.41	0.64	VAPEEHPTLL	1.87	−0.53	−0.81	1.14	ELPDGQVIT	1.14	−1.76	−0.72	1.69
LEKSYELPDGQVIT	1.27	−0.55	−0.43	0.82	GFAGDDAPRAVFPS	1.75	0.29	−0.86	0.71	VIDQDKSGFIE	1.13	0.37	−0.43	0.88
VETEKTEIQSALEEAEGTLEHEESKILR	1.00	**2.29**	−0.80	0.25	GAAGPAGPSGPRGPAGIA	1.62	0.16	−0.51	0.23	IIAPPERKYS	1.11	−0.59	−0.62	**3.43**
LDDLQAEEDKVNT	0.94	0.55	−1.02	1.17	RVAPEEHPTLL	1.53	0.36	−0.09	1.79	LEDQLSELK	1.06	1.36	−1.20	−0.22
LADWKQKYEEGQAELEGSLKEARS	0.74	0.70	−0.08	0.43	ELPDGQVITIG	1.34	−0.99	0.01	2.07	GVDNPGHPF	1.00	1.13	−0.49	0.08
VAPEEHPTL	0.72	-0.18	−0.07	1.04	KSYELPDGQVITIG	1.21	−0.78	−0.79	**2.64**	GADPEDVIVA	0.95	−1.29	−1.00	**3.10**
LDDVIQTGVDNPGHPFIMT	0.71	0.83	0.33	−0.02	LLPVLYPPVVEE	1.09	0.24	−1.23	1.33	AAGPAGPSGPRGPAG	0.75	−0.56	−0.75	0.55
LEDECSELKKDIDDLELT	0.71	1.84	−0.42	0.82	LGEQIDNL	1.03	**2.57**	−0.40	−0.40	AGDDAPRAVFPS	0.71	0.97	−1.33	1.23
ARIEELEEELEAERA	0.68	−0.65	−0.93	0.54	RGDSGPAGPPGEQGML	0.94	−1.16	−0.62	0.13	LDKNKDPLNDSVVQ	0.68	1.41	−0.79	1.18
LTEEMASQDESVAK	0.67	1.01	−0.17	0.11	GSAGPRGPSGNIGMPGMTGPQ	0.85	−1.75	−0.73	0.14	FAGDDAPRA	0.67	0.49	−0.69	1.15
IEELEEELEAERA	0.65	0.22	0.05	1.05	TIIDQNRDGIIS	0.76	0.70	−0.99	1.49	ELPDGQVITI	0.67	−0.93	0.00	**2.37**
LDFENEMAT	0.65	0.10	1.04	0.59	GEKLKGADPEDVIVA	0.74	−1.01	−0.41	**2.36**	KSYELPDGQVITIG	0.63	−0.78	−0.79	**2.64**
VDDIIQTGVDNPGHPFIMT	0.63	0.89	0.34	0.07	SGSAGKDGMSGLPGPSGPSGPRGR	0.67	−0.23	−1.12	0.32	GFNPPDLDIM	0.60	−0.75	−0.19	1.20
LKAGDSDGDGAIGVDEWA	0.61	0.26	−0.18	0.56	SGPGGPTGPSGM	0.67	−1.24	−0.74	−0.49	ELPDGQVI	0.60	−1.92	−0.85	0.68
LVQVQGEVDDSVQEARNAEEKAKKA	0.60	**2.64**	1.54	**2.42**	GEQIDNL	0.67	1.96	−1.14	0.44	NPPKYDKIEDM	0.55	−0.15	−0.70	1.79
ALEEAEGTLEHEESKLLR	0.58	1.83	−0.16	0.57	ELPDGQVI	0.66	−1.92	−0.85	0.68	KSYELPDGQVITI	0.53	−0.70	−0.82	**2.76**
IEELEEELEAERAAR	0.56	−0.70	0.82	0.31	KSYELPDGQVITI	0.65	−0.70	−0.82	**2.76**	KSYELPDGQVIT	0.51	−1.28	−1.17	1.13

**Table 4 marinedrugs-18-00599-t004:** Sequences, relative intensity (RI), and predicted free radical scavenging (FRS) and metal chelating activity (MCA) scores for the 20 peptides with highest relative intensity from enzymatically derived FPH from minced cod frame (MCF) and heated cod meat (HCM). DPPH scores >0.43 and MCA scores >0.3 are highlighted in bold providing the best cutoff between positive and negative predictions according to the models [77].

**MCF**											
**Neut.**	**Rel. Int. (%)**	**Score**		**Alc**	**Rel. Int. (%)**	**Score**		**Neut&Alc**	**Rel. Int. (%)**	**Score**	
		**FRS**	**MCA**			**FRS**	**MCA**			**FRS**	**MCA**
LQGEVEDLMVDVERANG	1.94	0.33	0.17	GFAGDDAPRAVFPS	2.30	**0.51**	0.25	IIAPPERKYS	3.49	0.40	0.23
LEQQVDDLEGSLEQEKK	1.92	0.38	0.19	RVAPEEHPTLL	2.13	**0.51**	0.26	GVDNPGHPF	2.00	**0.56**	0.28
IITNWDDMEK	1.78	0.38	0.19	AGDDAPRAVFPS	2.09	0.41	0.24	LDLL	1.75	0.35	0.28
LTKLEEAEKAADESERGMK	1.17	0.29	0.17	AGPAGPSGPRGPAGIA	1.56	**0.51**	0.21	VGPF	1.65	**0.47**	0.27
LEDQLSEIKAKSDENARQ	1.00	0.23	0.17	FLGM	1.34	**0.43**	0.28	VAPEEHPTL	1.59	**0.51**	0.25
LEKSYELPDGQVIT	0.92	**0.48**	0.23	KSYELPDGQVITIG	1.33	0.35	0.21	VLYPPVVEE	1.53	**0.52**	0.21
VAPEEHPTL	0.90	**0.51**	0.25	GAAGPAGPSGPRGPAGIA	1.30	**0.49**	0.20	GADPEDVIVA	1.43	0.32	0.20
IIDQNRDGIISKDDLRD	0.86	0.30	0.22	DIDIRKDLYAN	1.16	0.27	0.17	VIDQDKSGFIE	1.35	0.30	0.19
LDDLQAEEDKVNT	0.81	0.34	0.23	VAPEEHPTLL	1.12	**0.56**	0.26	AGDDAPRAVFPS	1.20	0.41	0.24
LEKTIDDLEDELYAQK	0.78	0.35	0.22	AGPSGPRGPAGIA	1.08	**0.47**	0.20	KSYELPDGQ	1.19	**0.44**	0.23
LKGTEDELDKYSEALKDAQEKLE	0.77	0.34	0.16	NWDDMEKIWHH	1.06	0.34	0.20	VAVL	1.17	0.33	0.23
LTEEMASQDESVAK	0.76	0.31	0.21	GQKDSYVGDEAQSKRGILTL	0.92	0.30	0.18	LFPE	1.17	**0.44**	0.30
LKGADPEDVIVAA	0.76	0.38	0.19	NWDDME	0.88	0.40	0.21	GLVL	1.16	0.37	0.24
LADWKQKYEEGQAELEGSLKEARS	0.71	0.30	0.15	LRVAPEEHPTL	0.85	**0.45**	0.25	LLEM	1.12	0.39	0.27
SKYETDAIQRTEELEESKKK	0.69	0.40	0.14	RGDSGPAGPPGEQGML	0.84	**0.51**	0.29	GERGEQGPGGPGGF	1.08	**0.52**	0.23
LKAGDSDGDGAIGVDEWAV	0.69	0.35	0.18	ELPDGQVITIG	0.74	0.31	0.20	LGVL	0.95	0.38	0.25
LTDAETKAF	0.65	0.33	0.24	SGSAGKDGMSGLPGPSGPSGPRGR	0.67	**0.55**	0.25	LDFENE	0.87	0.37	0.27
LKAGDSDGDGAIGVDEWAVLVKA	0.61	0.36	0.13	LDLL	0.67	0.35	0.28	IIDQNRDGIIS	0.83	0.29	0.19
VDDIIQTGVDNPGHPFIMT	0.59	0.42	0.20	LLPVLYPPVVEE	0.65	**0.64**	0.21	AGPAGPSGPRGPAG	0.81	**0.49**	0.22
VIDQDKSGFIEEDELKLF	0.57	0.36	0.19	SGPGGPTGPSGM	0.64	**0.50**	0.23	LLLS	0.79	0.34	0.29
**HCM**											
**Neut.**	**Rel. Int. (%)**	**FRS**	**MCA**	**Alc.**	**Rel. Int. (%)**	**FRS**	**MCA**	**Netu&Alc**	**Rel. Int. (%)**	**FRS**	**MCA**
LEQQVDDLEGSLEQEKK	2.38	0.38	0.19	AGDDAPRAVFPS	3.13	0.41	0.24	VAPEEHPTL	1.91	**0.51**	0.25
LQGEVEDLMVDVERANG	2.28	0.33	0.17	GPAGPSGPRGPAGIA	2.64	**0.49**	0.21	VLYPPVVEE	1.75	**0.52**	0.21
IITNWDDMEK	2.23	0.38	0.19	AGPAGPSGPRGPAGIA	1.90	**0.51**	0.21	IIDQNRDGIIS	1.45	0.29	0.19
VQHELEEAEERADIAETQVNK	1.46	0.33	0.15	VAPEEHPTLL	1.87	**0.56**	0.26	ELPDGQVIT	1.14	0.36	0.23
LEKSYELPDGQVIT	1.27	**0.48**	0.23	GFAGDDAPRAVFPS	1.75	**0.51**	0.25	VIDQDKSGFIE	1.13	0.30	0.19
VETEKTEIQSALEEAEGTLEHEESKILR	1.00	0.37	0.13	GAAGPAGPSGPRGPAGIA	1.62	**0.49**	0.20	IIAPPERKYS	1.11	0.40	0.23
LDDLQAEEDKVNT	0.94	0.34	0.23	FLGM	1.53	**0.43**	0.28	FLPM	1.07	**0.46**	**0.31**
LADWKQKYEEGQAELEGSLKEARS	0.74	0.30	0.15	RVAPEEHPTLL	1.42	**0.51**	0.26	LEDQLSELK	1.06	0.34	0.25
VAPEEHPTL	0.72	**0.51**	0.25	ELPDGQVITIG	1.34	0.31	0.20	GVDNPGHPF	1.00	**0.56**	0.28
LFQPSF	0.72	0.42	0.28	KSYELPDGQVITIG	1.21	0.35	0.21	GADPEDVIVA	0.95	0.32	0.20
LDDVIQTGVDNPGHPFIMT	0.71	0.41	0.21	LLPVLYPPVVEE	1.09	**0.64**	0.21	AAGPAGPSGPRGPAG	0.75	**0.49**	0.22
LEDECSELKKDIDDLELT	0.71	0.26	0.18	LGEQIDNL	1.03	0.35	0.22	AGDDAPRAVFPS	0.71	0.41	0.24
ARIEELEEELEAERA	0.68	0.36	0.20	NWDDME	0.94	0.40	0.21	LDKNKDPLNDSVVQ	0.68	0.29	0.27
LTEEMASQDESVAK	0.67	0.31	0.21	RGDSGPAGPPGEQGML	0.90	**0.51**	0.29	FAGDDAPRA	0.67	0.37	0.22
FLPM	0.66	**0.46**	**0.31**	FETF	0.85	0.40	0.25	ELPDGQVITI	0.67	0.33	0.20
IEELEEELEAERA	0.65	0.36	0.20	GSAGPRGPSGNIGMPGMTGPQ	0.84	**0.48**	0.23	KSYELPDGQVITIG	0.63	0.35	0.21
LDFENEMAT	0.65	0.37	0.25	TIIDQNRDGIIS	0.76	0.29	0.20	LDFENE	0.60	0.37	0.27
VDDIIQTGVDNPGHPFIMT	0.63	0.42	0.20	GEKLKGADPEDVIVA	0.70	0.38	0.17	GFNPPDLDIM	0.60	0.39	0.25
LKAGDSDGDGAIGVDEWA	0.61	0.35	0.19	SGSAGKDGMSGLPGPSGPSGPRGR	0.67	**0.55**	0.25	ELPDGQVI	0.60	0.36	0.25
LVQVQGEVDDSVQEARNAEEKAKKA	0.60	0.34	0.11	SGPGGPTGPSGM	0.67	**0.50**	0.23	LDLL	0.58	0.35	0.28

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
