# Peer review of "Biofunctionality of Enzymatically Derived Peptides from Codfish (Gadus morhua) Frame: Bulk In Vitro Properties, Quantitative Proteomics, and Bioinformatic Prediction"

_marinedrugs, 2020, doi:10.3390/md18120599_

Round 1

Reviewer 1 Report

This paper describes  chemical and biological characterization of peptides derived from Cod fish frame by enzymatic hydrolysis. The methods used and results obtained are solid and interesting, and accordingly, this paper deserves to be published in this journal.

It is better to show LC-MS (TIC) profiles of FPH samples to more easily understand the enzymatic hydrolysis results.

Author Response

We appreciate the positive feedback from the reviewer on our manuscript. Although we do recognize the background for the reviewer’s suggestion of including the MS1 TIC profile as means to better understand and visualize the hydrolysis and the differences between the hydrolysates. Our evaluation is that they bring very limited value to the manuscript. This is based on the fact that peptide separation is achieved using a RP-nLC column which separates peptides according to hydrophobicity. Had chromatography been done using a size exclusion column, the TIC would to a much larger degree reflect this issue and illustrate the degree of hydrolysis. Furthermore, as we in each hydrolysate identify roughly 4000-6000 peptides, this number is not reflected in the number of observable peaks in the MS1 TIC, as each peak can easily include a multitude of peptides with comparable hydrophobicity. As such, the TIC does not provide an illustrative example of the complexity in our opinion. Nevertheless, we do recognize that the TIC does show that the sample composition changes as a function of the applied protease, although the directly extractable information here from is sparse. As such, we have included an example of such a plot (an overlay of the TIC for the three hydrolysates from HCM) below to illustrate our view of the suggested addition. We would also like to emphasize, that should the reviewer maintain the wish to have the data included, we can quickly plot this and append it to the supporting information and refer appropriately in the manuscript.

Reviewer 2 Report

The authors investigated peptides from Codfish that are important for the food industry. A novel bioinformatics approach was presented for calculating the peptide properties. It highlights the prospects of applying proteomics and bioinformatics for hydrolase characterization in food protein science.

The study characterizes the bulk emulsifying, foaming and antioxidative properties of hydrolases. Interesting histograms are presented in the results showing the distribution of the peptide chain length, molecular weight and charge.

In general the article is well written and deserves publication. I wish to suggest the following points.

  1. The abstract could be improved to emphasize the novelty of the paper related to the bioinformatics approach.
  2. A recent paper using similar bioinformatics approach should be appended to the list of references:

Prediction of Amphiphilic Cell-Penetrating Peptide Building Blocks from Protein-Derived Amino Acid Sequences for Engineering of Drug Delivery Nanoassemblies, Journal of Physical Chemistry B, 2020, 124(20), pp. 4069-4078.

Author Response

We would like to thank the reviewer for the very positive feedback on our manuscript. As for the specific points raised by the reviewer, we have the following replies:

  1. We agree with the suggestion and have revised the abstract accordingly to highlight and emphasize the novelty of the applied methodology.
  2. Although the suggested reference relates to cell penetrating peptides and does not relate to emulsifying properties, we do see the point of the reviewer. As the paper does cover bioinformatic prediction of peptide building block used for the design amphiphilic peptides (axial amphiphilicity to some extent comparable to the γ-peptides predicted in this work), we have included the reference and briefly presented and discussed it (L.312ff)